# AI-Based Risk Score from Tumour-Infiltrating Lymphocyte Predicts Locoregional-Free Survival in Nasopharyngeal Carcinoma

**DOI:** 10.3390/cancers15245789

**Published:** 2023-12-10

**Authors:** Made Satria Wibawa, Jia-Yu Zhou, Ruoyu Wang, Ying-Ying Huang, Zejiang Zhan, Xi Chen, Xing Lv, Lawrence S. Young, Nasir Rajpoot

**Affiliations:** 1Tissue Image Analytics Centre, Department of Computer Science, University of Warwick, Coventry CV4 7AL, UK; made-satria.wibawa@warwick.ac.uk (M.S.W.); ruoyu.wang.2@warwick.ac.uk (R.W.); 2State Key Laboratory of Oncology in South China, Guangdong Key Laboratory of Nasopharyngeal Carcinoma Diagnosis and Therapy, Guangdong Provincial Clinical Research Center for Cancer, Sun Yat-Sen University Cancer Center, Guangzhou 510060, China; zhoujy@sysucc.org.cn (J.-Y.Z.); huangyy4@sysucc.org.cn (Y.-Y.H.); zhanzj@sysucc.org.cn (Z.Z.); chenxi15@sysucc.org.cn (X.C.); lvxing@sysucc.org.cn (X.L.); 3Department of Nasopharyngeal Carcinoma, Sun Yat-Sen University Cancer Center, Guangzhou 510060, China; 4Warwick Medical School, University of Warwick, Coventry CV4 7AL, UK; l.s.young@warwick.ac.uk; 5The Alan Turing Institute, London NW1 2DB, UK

**Keywords:** nasopharyngeal carcinoma, tumour-infiltrating lymphocytes, locoregional recurrence, computational pathology, artificial intelligence

## Abstract

**Simple Summary:**

Plasma Epstein–Barr virus (EBV) DNA is an important prognostic marker for nasopharyngeal carcinoma (NPC). However, EBV DNA is less sensitive to locoregional recurrence compared to distant metastasis in NPC. Numerous findings suggest that the presence of tumour-infiltrating lymphocytes (TILs) is associated with NPC prognosis. Nevertheless, NPC is characterised by the presence of abundant TILs. This study aims to generate TIL scores in NPC from H&E-stained tissue slide images for NPC prognosis. We employed artificial intelligence and deep learning-based method for generating TIL score. Our results indicate that our methods have strong prognostic value compared to the EBV DNA copies in locoregional recurrence cases.

**Abstract:**

Background: Locoregional recurrence of nasopharyngeal carcinoma (NPC) occurs in 10% to 50% of cases following primary treatment. However, the current main prognostic markers for NPC, both stage and plasma Epstein–Barr virus DNA, are not sensitive to locoregional recurrence. Methods: We gathered 385 whole-slide images (WSIs) from haematoxylin and eosin (H&E)-stained NPC sections (*n* = 367 cases), which were collected from Sun Yat-sen University Cancer Centre. We developed a deep learning algorithm to detect tumour nuclei and lymphocyte nuclei in WSIs, followed by density-based clustering to quantify the tumour-infiltrating lymphocytes (TILs) into 12 scores. The Random Survival Forest model was then trained on the TILs to generate risk score. Results: Based on Kaplan–Meier analysis, the proposed methods were able to stratify low- and high-risk NPC cases in a validation set of locoregional recurrence with a statically significant result (*p* < 0.001). This finding was also found in distant metastasis-free survival (*p* < 0.001), progression-free survival (*p* < 0.001), and regional recurrence-free survival (*p* < 0.05). Furthermore, in both univariate analysis (HR: 1.58, CI: 1.13–2.19, *p* < 0.05) and multivariate analysis (HR:1.59, CI: 1.11–2.28, *p* < 0.05), we also found that our methods demonstrated a strong prognostic value for locoregional recurrence. Conclusion: The proposed novel digital markers could potentially be utilised to assist treatment decisions in cases of NPC.

## 1. Introduction

Nasopharyngeal carcinoma (NPC) is a rare malignancy that originates from the epithelial cells in the nasopharynx, with an epicentre in the Fossa of Rosenmuller, a pharyngeal recess within the nasal cavity. This location allows the tumour to spread through various routes [1,2,3]. It has a unique aetiology involving environmental factors, genetics and infection with the Epstein–Barr virus (EBV) [4,5,6]. The primary corrective treatment for NPC is radiation therapy (RT), which may be complemented with either concurrent, induction or adjuvant chemotherapy in advanced cases [7,8]. Clinically, advanced-stage nasopharyngeal carcinoma is typically categorized as locally advanced nasopharyngeal carcinoma (LA NPC), encompassing stage III (T3N0–2M0, T0–2N2M0) and stage IVa (T4 or N3M0) according to the UICC/AJCC Clinical Staging Criteria (8th Edition).

Distant metastases are the main cause of treatment failure. Nonetheless, locoregional recurrence occurs for 10% to 50% of the cases following primary treatment at 5 years [9,10]. This can lead to a higher risk of further recurrence or metastasis to other parts of the body. Early detection and treatment of locoregional recurrence can prevent further tumour spread and improve NPC prognosis [11].

The TNM staging system is the current primary factor for determining treatment decisions and prognosis for patients with NPC. However, despite NPC patients receiving similar treatment at the same stage, their outcomes may vary [12]. The level of circulating cell-free EBV DNA in the plasma of NPC patients has been shown to be an important prognostic marker and can aid in determining appropriate treatment [13,14]. However, EBV DNA is not as sensitive to locoregional recurrence as it is to distance metastasis [15]. It has been demonstrated that as much as 40% of local recurrence is not associated with an increase in plasma EBV DNA [16].

NPC is characterised by the presence of abundant tumour-infiltrating lymphocytes (TILs) [17]. There has been a growing interest in the potential prognostic value of TILs in various types of cancer, including NPC [18]. The presence of TILs within the tumour microenvironment has been associated with better clinical outcome in several malignancies. In NPC, the relationship between TILs and tumour progression has been the subject of numerous studies. Several investigations have reported a positive correlation between the density of TILs and favourable clinical outcomes in NPC patients. In particular, the presence of TILs has been associated with improved overall survival, disease-free survival, and distant metastasis-free survival [19,20,21,22]. These findings suggest that TILs may play a critical role in the host immune response against NPC and may serve as a prognostic biomarker for the disease. Despite this potential, it is imperative to reach a consensus on the definition and enumeration of TILs [23] to circumvent subjectivity and intra-observer variability.

Similar to other cancers, NPC diagnosis and prognosis involves a multifaceted approach. This includes symptom checking, physical exams, endoscopies, advanced imaging (MRI, CT, PET), and blood tests. To further validate the presence of tumours and refine the diagnosis, pathology tests are conducted. This involves a careful examination of tissue samples to confirm the existence of tumours and their grading. Simultaneously, radiological assessments are employed for cancer staging, aiding in determining the extent and severity of the disease. The culmination of these diagnostic efforts is a labour- and time-intensive process that extends over the course of more than one week.

Artificial intelligence (AI) and computational pathology in cancer treatment has two primary goals: automating portions of the routine clinical workflow and gaining new insights using data from that same workflow [24]. The emergence of AI and computational pathology in the diagnosis and prognosis of cancer, including NPC, is beginning to deliver promising results in several tasks [25,26,27]. For instance, AI algorithms have been shown to perform better than human observers in detecting nasopharyngeal mass in endoscopic images [28]. As another example in NPC diagnosis using whole-slide images (WSIs), AI algorithms have been shown to outperform junior and intermediate pathologists [29]. AI has demonstrated its potential in NPC prognosis tasks, achieving high accuracy by utilising various data modalities, such as clinicopathological data, magnetic resonance imaging (MRI), (positron emission tomography) PET, and WSIs [30,31,32,33]. Compared to other data modalities such as MRI and PET, WSIs offer a comprehensive view of cellular and tissue structures. However, to the best of our knowledge, there are no existing studies examining AI for the detailed quantification of TILs in NPC WSIs.

The aim of this study is to categorise patients with locally advanced nasopharyngeal carcinoma into high- and low-risk groups using an AI-generated risk score derived from histology images. We examine the potential of AI and computational pathology to develop a novel digital TIL-based score for NPC (digital NPC-TILs) prognosis in terms of locoregional recurrence-free survival (LRFS). Aside from the LRFS, we also examine distant metastasis-free survival (DMFS), regional recurrence-free survival (RRFS), and progression-free survival (PFS). By adopting AI and computational pathology, our methods offer an objective assessment of TILs from widely used H&E-stained histological images. We conduct detailed analyses and show that our novel digital TIL score has prognostic value in terms of LRFS as well as the other survival endpoints. Our study offers insights into the correlation between TILs, the tumour microenvironment, and patient survival in NPC. The proposed method holds potential for aiding decision making in NPC treatment.

## 2. Materials and Methods

### 2.1. Data Collection

A total of 396 whole-slide images (WSIs) for haematoxylin and eosin (H&E)-stained NPC sections (*n* = 378 cases) were collected from Sun Yat-sen University Cancer Centre. All WSIs were scanned with 40× magnification at a resolution of 0.255 microns per pixel (mpp) using an Aperio scanner. Among the total of 378 cases in our cohort, 11 cases only have WSIs without any accompanying clinical information, including survival endpoints. This resulted in a final dataset of 367 cases with a total of 385 WSIs. For cases with multiple WSIs, features were aggregated as the average value. Approval was obtained from the ethics committee of the Sun Yat-sen University Cancer Centre (B2023-381-01).

The clinical data include age, sex, primary tumour (T), regional lymph node (N), stage group, and plasma EBV DNA data. M is not included in this study as our focus is on predicting the prognosis of patients with locally recurrent NPC, excluding those with metastasis to other organs. Additionally, the TNM staging status was acquired at the beginning of the study when patients exhibited early symptoms, and thus, metastasis had not yet developed. As in a previous study [34], plasma EBV DNA levels were quantified using quantitative polymerase chain reaction targeting the BAMHI-W region of the EBV genome prior to treatment. The result was reported as the concentration of EBV genome copies per millilitre of plasma. All patients in the cohort received both chemotherapy and radiotherapy.

### 2.2. Clinicopathological Features

The median follow-up duration in this cohort was 36 months, with a range duration of 1–79 months. The survival data were right censored by limiting the maximum observed time to 60 months (5 years). The average age of all patients was 45 with 70.84% being male. The percentage of events in LRFS, DMFS, PFS, and RRFS were 5%, 12%, 17%, and 2%, respectively. We stratified patients into high- and low-EBV DNA groups for prognosis analysis using 4000 copies/mL as a threshold (low: ≤4000 copies/mL, high: >4000 copies/mL). This threshold was chosen to align with the latest authoritative studies [35,36]. The cohort consists of 167 high EBV DNA copies cases and 200 low EBV DNA copies. The TNM stage in the cohort was measured using the eighth edition of the International Union Against Cancer (UICC) and the American Joint Committee on Cancer (AJCC) staging system. The majority of patients were in stage 3 based on the TNM staging system (Table 1).

### 2.3. Tumour Nuclei Detection and Clustering

Our overall analysis pipeline, as shown in Figure 1, consists of four main steps, namely tumour nuclei detection and clustering, digital TIL quantification, risk score calculation and survival analysis. We detected, segmented, and classified nuclei in WSIs at 40× magnification using the HoverNet model [37], which was trained using PanNuke dataset [38].

In order to correct the misclassification of the nuclei, we trained a deep learning model that we term as MorphResNet that incorporates feature fusion from deep features of the Residual Neural Network (ResNet [39]) and morphology features of nuclei. We utilised features from the penultimate layer of ResNet50, which was pretrained on the ImageNet [40] dataset. We chose ResNet model due to its widespread implementation, as it not only saves computational costs and time, but also maintains predictive power without degradation [41,42]. The morphology features consist of eccentricity, convex area, contour area, equivalent diameter, major axis length, minor axis length, perimeter, solidity, orientation, radius, and bounding box area. The complete model architecture can be found in Appendix A.

To construct nuclei dataset, we amalgamated data from PanNuke and MoNuSAC [43] to create nuclei datasets. We selectively included data labelled as “epithelial” and “inflammatory” from MoNuSAC. Each nuclei images in datasets were extracted and centred to a size of 64 × 64 pixels, with additional padding for those located near the image edge. Total number of nuclei images in the final dataset was 219,934. We augmented our data using the Macenko [44] method to ensure the model’s generalisability in the variance of image staining. The data were split into 80-10-10 ratio for training, validation, and testing sets. We trained all layers in model using Adam optimizer for 50 epochs. The model achieved an accuracy of 0.921 and F1-score of 0.919, respectively in the test set. The model’s performance was also assessed qualitatively and approved by the medical personnel involved in the study.

We clustered tumour nuclei based on their density with the Density-Based Spatial Clustering of Applications with Noise (DBSCAN) algorithm [45]. Furthermore, the tumour cluster was expanded by 60 pixels or 15 microns, resulting in a new cluster with a larger area. We set the maximum distance to 200 and the minimum number of samples to 10 in DBSCAN. All the parameters were determined empirically. These parameters allow us to avoid overlap among clusters. The initial cluster was referred to as the inner tumour cluster (in the midst of tumour clusters), while the expanded region was designated as the outer tumour cluster.

### 2.4. Digital TILs and Tumour Morphology

Several studies have shown that tumour microenvironment (TME) correlates with cancer survival. TME consists of stromal cells such as fibroblasts, endothelial cells, and immune cells (lymphocytes) around the tumour region. In this study, our focus was on tumour and lymphocytic regions and their relation to survival. We formulated digitised TIL scores based on nuclei in H&E images. All lymphocyte nuclei in and around tumour clusters were defined as tumour-infiltrating lymphocytes (TILs).

We term the tumour cells and TILs located in the inner tumour cluster as intratumoural tumours cells and intratumoural TILs (Figure 2). In contrast, tumours and TIL nuclei located in the outer tumour cluster were defined as stromal tumour cells and stromal. The two types of tumour nuclei and TILs are illustrated in Figure 1. By employing measurement of the count and the area of the tumour nuclei and TILs with their respective cluster, we generated 12 digital TIL features. We called this set of features as digital NPC-TILs. The complete list of the digital NPC-TIL features and their formulae can be seen in Appendix A. Apart from that, we also examined the tumour morphology in NPC survival. We calculated 9 morphology features from tumour nuclei inside the cluster area. The tumour nuclei in the expanded region of the cluster were not counted for the morphology features measurement. Next, we calculate the mean of the morphology features for each patient and utilise it for downstream analysis. The complete list of morphology features utilised in this study can be seen in Appendix A.

### 2.5. Survival Analysis

We determined locoregional recurrence-free survival (LRFS) as the primary endpoint in our survival analysis. We also examined additional endpoints such as DMFS, RRFS, and PFS. Patients without an event at the last date of the study period were labelled as a censored event. Python 3.7 (Python Software Foundation, www.python.org, accessed on 6 February 2023) programming language was used in the whole pipeline, from nuclei detection and classification to digital TIL scoring and statistical analyses.

In this study, we sought to develop digital AI-based biomarkers that carry prognostic value in NPC. We employed four sets of features in our survival analyses, namely digital TIL scores, clinicopathological data (age, sex, TNM staging, and EBV DNA levels), tumour nuclei morphology, and a combination of digital NPC-TIL scores with tumour nuclei morphology. We trained the Random Survival Forest (RSF) model with each feature set on three-fold cross-validation. We performed a grid search to determine the optimal parameters for RSF. To assess risk score prediction from the model, we leverage Harrell’s C-index (concordance index) [46] from the function concordance_index in PySurvival [47] Python package. C-index evaluates the probability of concordance between the predicted survival outcome and the actual survival outcome [48].

Subsequently, we assess the prognostic value of risk scores from the RSF model to stratify patients into low- and high-risk groups. Patients with risk scores equal to or lower than the cut-off value were considered low-risk and vice versa. The cut-off value was determined in the discovery set and calculated using Cutoff Finder [49]. We used Kaplan–Meier (KM) curves analysis with the log-rank test to evaluate the proposed biomarker in NPC prognosis in the validation set.

Furthermore, we carried out univariate and multivariate analyses to evaluate the relationship between risk factors and survival outcomes using the Cox proportional hazards (Cox PH) model. We utilised hazard ratio (HR), C-index, 95% confidence interval (CI), and *p*-value in the lifelines Python package (https://lifelines.readthedocs.io/ accessed on 6 March 2023) to appraise the Cox PH model. All statistical analyses were conducted using a two-tailed test, and a *p*-value lower than 0.05 was considered significant.

To conduct prognostic value assessment and univariate and multivariate analyses, we partitioned the data into discovery and validation sets. The validation set was set to 30% of the data, and the data were stratified based on the event of the corresponding endpoint.

## 3. Results

As mentioned above, we compared four sets of features in NPC prognosis: clinicopathological, digital NPC-TILs, tumour morphology, and the combination of digital NPC-TILs and tumour morphology. Three-fold cross-validation experiments were conducted to assess the generalisability of the proposed AI models. In cross-validation, data were shuffled randomly and stratified into three groups/folds. These folds were utilised for the discovery and validation of RSF models. For each validation fold, we calculated the risk score from RSF based on digital TIL features and their values as obtained from the discovery fold.

### 3.1. Patient Stratification with Risk Score of Digital NPC-TILs

We determined locoregional recurrence-free survival (LRFS) as the primary endpoint in our survival analysis. We also examined additional endpoints such as DMFS, RRFS, and PFS. Patients without an event at the last date of the study period were labelled as a censored event. Python 3.7 programming language was used in the whole pipeline, from nuclei detection and classification to digital TIL scoring and statistical analyses. The cut-off for stratifying patients in LRFS, DMFS, PFS, and RRFS were 4.52, 7.42, 28.20, and 3.86, respectively.

As can be observed in Figure 3, the proposed digital NPC-TIL score was able to stratify low- and high-risk NPC cases in LRFS with a statistically significant result (*p* = $8.13\times 10^{-5}$). We also obtain a similar result in other survival endpoints, such as DMFS (*p* = $5.83\times 10^{-4}$), PFS (*p* = $6.24\times 10^{-4}$), and RRFS (*p* = $5.67\times 10^{-3}$).

### 3.2. Correlation Analysis of Predicted Risk Scores and Time to Event

We examined the performance of our digital NPC-TIL score in survival prediction with the RSF model and compared it with other feature sets. Our digital NPC-TIL score alone performed better compared to other feature sets in LRFS. On the discovery set, digital NPC-TIL score achieved an average C-index of 0.923 and 0.785 on the validation set. Meanwhile, other feature sets achieved lower C-index values (Table 2). Additionally, digital NPC-TIL score also delivered the highest C-index in other survival endpoints. It achieved a C-index of 0.780 in RRFS and 0.667 in PFS. In the case of DMFS, the digital NPC-TIL score achieved a C-index of 0.645, which was lower than the C-index of 0.670 associated with the clinical feature. For the complete C-index results, please refer to Appendix A.

### 3.3. Univariate and Multivariate Analyses

We inspected the relationship between a single risk factor of each clinicopathological data and the risk score of digital TILs with LRFS using the Cox PH regression model (Table 3). In the univariate analysis, we found that only digital NPC-TIL score has prognostic value for LRFS (*p* < 0.05) with HR = 1.58 (95% CI = 1.13–2.19).

We also found that digital NPC-TIL score has a strong prognostic value in DMFS and PFS. In DMFS, the EBV DNA copies (HR: 3.88, 95% CI: 1.21–12.47, *p* < 0.05) and the proposed digital NPC-TIL score (HR: 1.36, 95% CI: 1.14–1.16, *p* < 0.001) were significantly correlated with survival outcome.

In PFS, the stage N = 3 (HR:6.7, 95% CI: 1.45–30.88, *p* < 0.05) and digital NPC-TIL score (HR: 1.08, 95% CI: 1.03–1.14, *p* < 0.005) were also significantly correlated with PFS outcome. In contrast, the digital NPC-TIL score was not significantly correlated with RRFS outcome (HR: 1.65, 95% CI: 0.85–3.21) with a *p*-value of 0.1388. In addition, none of the other features were significantly correlated in the validation set of RRFS. The univariate analysis results on DMFS, RRFS, and PFS can be seen in Appendix A.

The result of the multivariate analysis on LRFS is shown in Table 4. In the multivariate analysis, we found that only the risk score of digital TILs has a prognostic value in NPC (*p* < 0.05, HR = 1.59, 95% CI = 1.11–2.28). Digital NPC-TIL score also has prognostic value in DMFS (*p* < 0.005, HR = 1.35, 95% CI = 1.10–1.67) and PFS (*p* < 0.005, HR = 1.08, 95% CI = 1.02–1.14). Apart from digital NPC-TILs, only gender covariate has prognostic value in DMFS. The complete list of multivariate analyses can be found in Appendix A.

### 3.4. Digital NPC-TIL Scores in Low-Risk and High-Risk Groups

We conducted further analysis in the validation set to investigate how digital NPC-TIL score differs in low- and high-risk group. We performed either *t*-test or a Mann–Whitney U test, depending on the data distribution, to compare the mean between the two groups. As depicted in Figure 4, the low-risk cases have higher mean in 12 digital NPC-TIL features compared to the high-risk group in LRFS with statistically significant value (*p* < 0.05).

The difference in digital NPC-TILs between low- and high-risk groups was evident upon examining the WSIs; see for instance, the TILs ratio as shown in Figure 5. Red objects in the figure are cancer cell nuclei and green objects are lymphocytes nuclei. As can be seen, the quantity of lymphocytes in cases with good prognosis is significantly higher compared to those cases with poor prognosis. In this particular case, there were almost no lymphocytic infiltrates in the tumour cluster.

### 3.5. Tumour Nuclear Morphology Analysis

We also examined how tumour morphology differs in low- and high-risk groups of LRFS. For this purpose, we extracted nine morphological features from all tumour nuclei in the midst of the tumour clusters. We investigated the size of the tumour nuclei and the shape irregularities of tumour nuclei; higher values indicate larger tumour nuclei exhibiting increased irregularity in shape. We performed the same method as shown in Figure 4 to test for statistically significant difference between two groups. The result of this analysis can be observed in Figure 6. The features in the first two rows in Figure 6 represent tumour cell sizes, while the rest features represent the irregularity in shape. All of the nine morphology features in the low-risk group have a lower mean compared to the high-risk group. Excluding solidity and eccentricity, all the features were found to have a statistically significant difference (*p* < 0.05).

### 3.6. Model Interpretability Analysis

Similar to other tree-based algorithms, RSF provides model interpretability via feature importance. Utilising the feature importance of RSF, we analysed the importance of each of the 12 digital TIL scores in terms of their contribution to the final digital NPC-TIL score in the validation set. As shown in Figure 7, the top 3 most important features in LRFS were stromal TIL density tumour, stromal TIL ratio, and TIL density cluster. Stromal TIL features (both tumour and ratio) refer to lymphocytes that are located at the outer edge of tumour clusters. Furthermore, the feature with the highest occurrence among the top three features across all survival endpoints (Appendix A) was the TIL density cluster, which captures information regarding the number of TILs within and around the tumour cluster.

## 4. Discussion

With current limitations of the TNM staging system and clinical data [50,51,52], there is increasing interest in immune features of the tumour microenvironment (TME) in NPC prognosis [22,53]. Despite the promising results of immune features of the TME (such as TILs) in other types of cancers, there is a dearth of literature in exploring immune features of the TME in NPCs. In addition, most immune features are subjectively assessed and are susceptible to inter- and intra-observer variation [54].

To address this issue, this study presents a novel method to automate the scoring of TILs based on H&E WSIs in NPC using artificial intelligence and computer vision. We developed 12 digital TIL features and combined them into a single NPC-TIL score. Our analysis suggests that this score is an independent prognostic factor for NPC in terms of LRFS and other survival outcomes in our cohort.

We captured TIL information in three locations: inner tumour cluster (in the midst of tumour cluster), outer tumour cluster (in the expanded area of tumour cluster), and in both of them. Each location encompasses four sets of digital NPC TILs. TILs extracted from the inner tumour cluster were given the prefix “Intratumoural” as they describe lymphocytes inside the high-density tumour area. Furthermore, TILs extracted from the outer tumour cluster were given the prefix “Stromal” as they were found around the tumour cluster.

The first set of digital NPC TILs provides a distribution of the number of lymphocyte cells with the number of tumour cells, which were named TIL ratio, intratumoural TIL ratio, and stromal ratio. The second set provides insights into the relationship between the number of lymphocyte cells and the total area occupied by tumour cells, and these were named TIL density cluster, intratumoural TIL density cluster, and stromal TIL density cluster. The third set compares the number of lymphocyte cells with the total area of tumour cells, named TIL density tumour, intratumoural TIL density tumour, and stromal TIL density tumour. Lastly, the fourth set delves into the spatial aspect by considering the total area covered by lymphocyte cells in relation to the total area of tumour cells, which were named the TIL area ratio, intratumoural TIL area ratio, and stromal TIL area ratio. Adopting these four feature sets across three distinct locations enhances our ability to characterize not only the quantity, but also the spatial distribution and density of TILs, providing a comprehensive understanding of their role in the intricate interplay within the tumour microenvironment.

It is known that EBV DNA is an important biomarker for distant metastasis in NPC [55,56]. However, this marker is known to be more sensitive to distant metastasis than to locoregional recurrence [57,58]. In our univariate and multivariate analyses with Cox-PH modelling, we showed that our digital NPC-TIL score is a strong prognostic factor in the LRFS of NPC. A stratification of low- and high-risk in locoregional recurrence NPC by employing this risk score yields a statistically significant result (*p* < 0.001).

The aim of this study is to stratify high- and low-risk groups among patients with locally advanced nasopharyngeal carcinoma through an AI-based risk score from histology images. For low-risk nasopharyngeal carcinoma patients, a “de-intensification” treatment strategy could be implemented to enhance their quality of life. Simultaneously, for high-risk locally advanced nasopharyngeal carcinoma cases, additional therapeutic measures, such as maintenance treatment with capecitabine following induction chemotherapy or incorporating full-course immunotherapy, could be considered.

We explored digital NPC-TIL scores on NPC prognosis in LRFS. As can be seen in Figure 4, digital NPC-TIL scores with good prognosis are significantly higher compared to those cases with poor prognosis. Our tumour nuclear morphology analysis suggested that digital NPC-TIL scores can capture nuclear aberration in local recurrence cases. In the poor prognosis cases, the tumour nuclei tend to have larger size and more irregular shape compared to tumour nuclei in good prognosis cases. It is known that nuclear tumour aberration is a hallmark feature of many tumours [59].

By examining model interpretability in our RSF model, we found that TIL density cluster is the most common feature in digital NPC-TILs with the highest importance across all survival endpoints. This feature captures information regarding the number of TILs within and around the tumour cluster pointing to immune response within the tumour microenvironment and confirming that the presence and quantity of TILs have implications for patient prognosis in NPC.

## 5. Conclusions

We have introduced a novel AI-based method that utilises routine H&E histology images and developed a set of novel digital TIL features for NPC prognosis. To the best of our knowledge, this is the first study to automate TIL scores in H&E images of NPC and to show potential prognostic markers which can help to assess patient treatment. The study provides important insights into the potential digital immune markers in NPC prognosis. The digital NPC-TIL scores could potentially be utilised to assist treatment decisions and improve patient outcomes in cases of local recurrence. Despite the promising findings about digital TILs in this study, our samples were derived solely from a single cohort. Furthermore, the absence of information on important factors such as lifestyle and comorbidities represent another constraint. Large-scale validation on multi-centric cohorts is required before the findings of this study can be implemented into clinical practice.

## Figures and Tables

**Figure 1 cancers-15-05789-f001:**
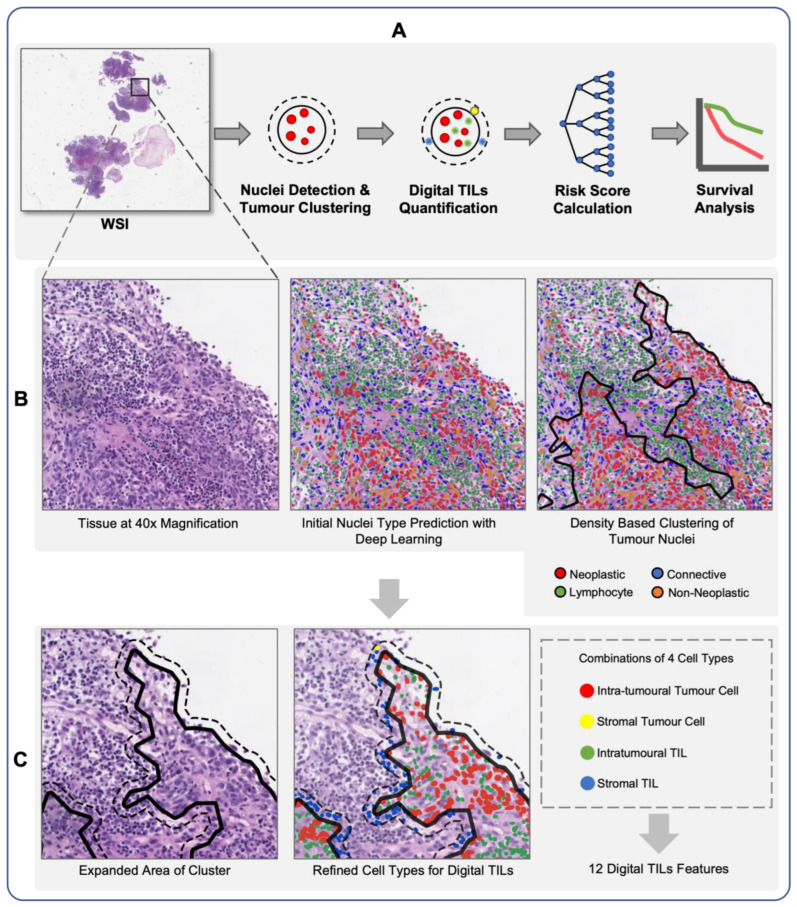
Illustration of the main workflow (**A**): nuclei detection at 40× magnification, followed by tumour nuclei clustering with density-based method, which resulted in inner tumour cluster (**B**); quantification of digital TILs (**C**).

**Figure 2 cancers-15-05789-f002:**
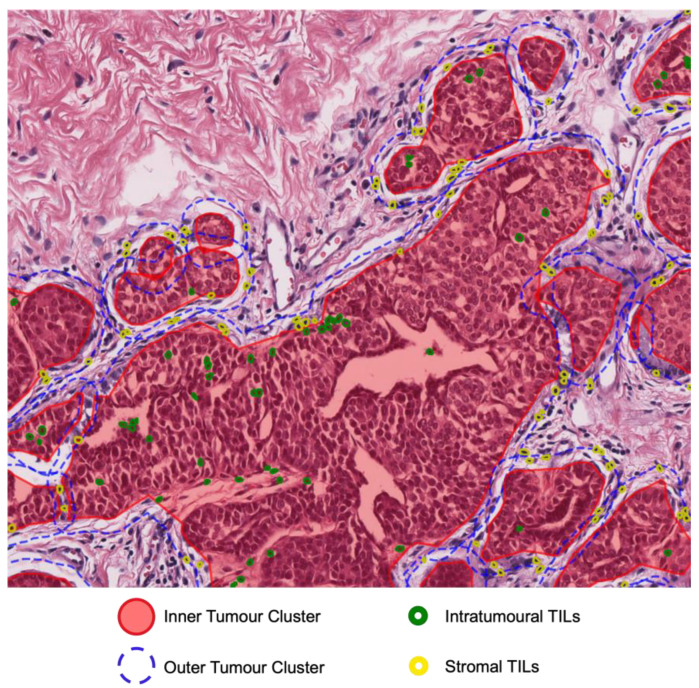
Main components of digital NPC-TILs. Inner and outer tumour cluster. Intratumoural and stromal TILs.

**Figure 3 cancers-15-05789-f003:**
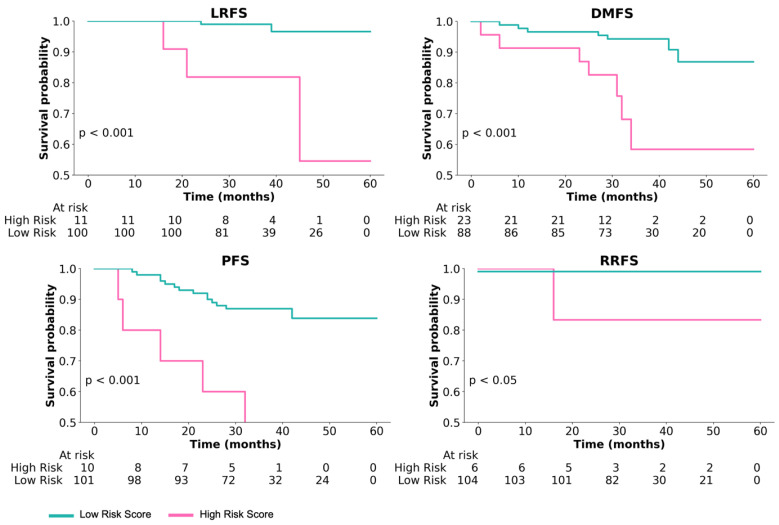
Kaplan–Meier curves for low- and high-risk groups based on the risk score of digital NPC-TILs.

**Figure 4 cancers-15-05789-f004:**
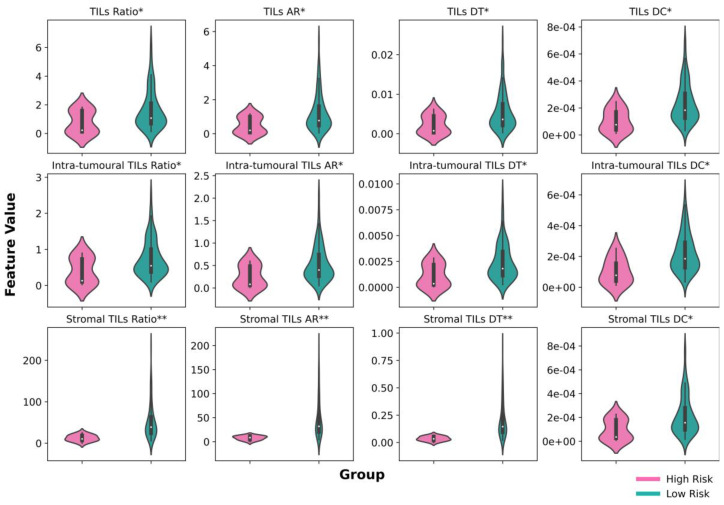
Comparison of digital NPC-TIL features on the low- and high-risk groups in LRFS. The * sign above the feature name denotes a statistically significant difference between low- and high-risk groups (*: *p* < 0.05; **: *p* < 0.001). AR = Area Ratio, DT = Density Tumour, DC = Density Cluster.

**Figure 5 cancers-15-05789-f005:**
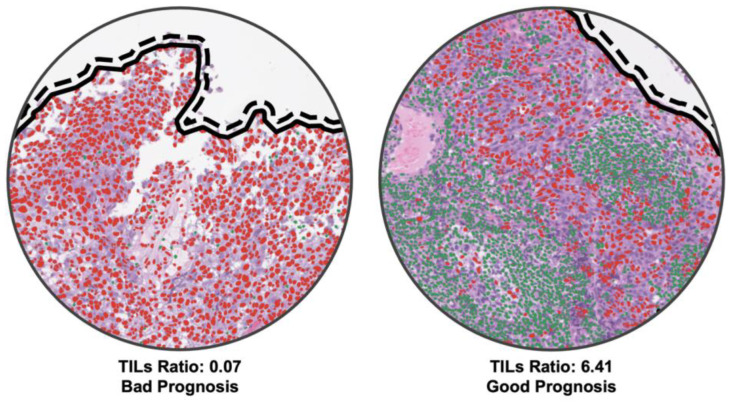
Sample high-power field images of low digital NPC-TIL score with bad prognosis (**Left**) and high digital NPC-TIL score with good prognosis (**Right**).

**Figure 6 cancers-15-05789-f006:**
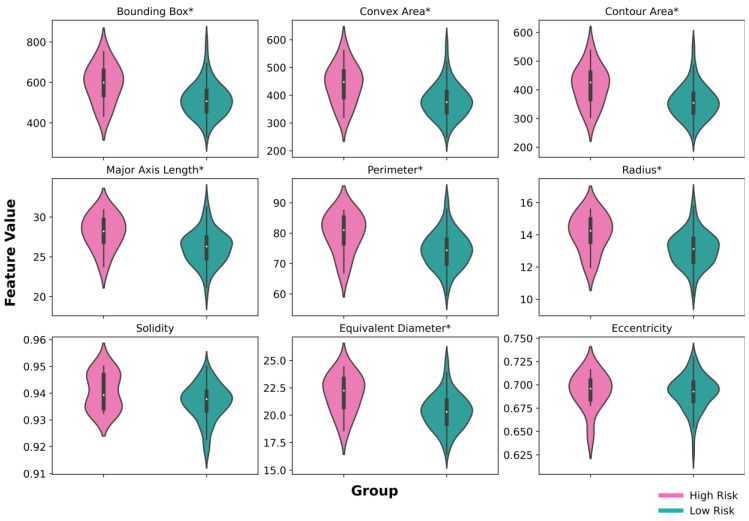
Tumour cell morphology comparison in low- and high-risk groups of LRFS. The * sign above the feature name denotes a statistically significant difference between low- and high-risk groups (*p* < 0.05).

**Figure 7 cancers-15-05789-f007:**
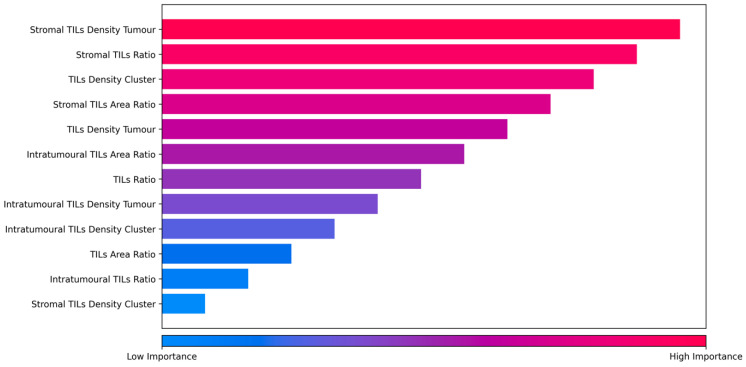
Rank of feature importance in risk score of LRFS.

**Table 1 cancers-15-05789-t001:** Summary of clinicopathological features in the cohort.

Variable	Sub-Variable	Count	% or Mean (SD)
Age		367	45.64 (11.30)
Sex	Male	260	70.84%
	Female	107	29.16%
T	T1	30	8.17%
	T2	49	13.35%
	T3	198	53.95%
	T4	90	24.52%
N	N0	36	9.81%
	N1	144	39.24%
	N2	127	34.60%
	N3	60	16.35%
Stage	I	8	2.18%
	II	30	8.17%
	III	194	52.36%
	IV	135	36.78%
EBV DNA copies	≤4000	200	54.50%
	>4000	167	45.50%

**Table 2 cancers-15-05789-t002:** Cross-validation results on LRFS prediction using RSF.

Features	Discovery Set C-Index (Mean ± SD)	Validation Set C-Index (Mean ± SD)
Clinical	0.873 ± 0.007	0.644 ± 0.124
Digital NPC-TILs	0.923 ± 0.024	0.785 ± 0.066
TM *	0.908 ± 0.017	0.624 ± 0.031
Digital NPC-TILs and Clinical	0.932 ± 0.027	0.670 ± 0.142
Digital NPC-TILs and TM *	0.936 ± 0.020	0.679 ± 0.120
Clinical and TM *	0.939 ± 0.010	0.629 ± 0.086
All Features	0.959 ± 0.008	0.689 ± 0.132

* TM: Tumour morphology.

**Table 3 cancers-15-05789-t003:** Univariate analysis of risk factors on LRFS.

Covariate	Sub-Covariate	HR	Lower HR 95%	Upper HR 95%	*p*-Values
Age		1.03	0.96	1.11	0.3566
Gender	Female	references			
	Male	1.64	0.25	10.70	0.6066
T	1	references			
	2	0.38	0.01	19.01	0.6273
	3	2.89	0.29	28.75	0.3661
	4	1.58	0.12	20.92	0.7291
N	0	references			
	1	0.77	0.10	5.79	0.8018
	2	0.44	0.05	3.82	0.4546
	3	0.76	0.07	7.88	0.8145
Stage	I	references			
	II	0.32	0	39.50	0.6425
	III	1.50	0.12	18.11	0.7514
	IV	1.36	0.11	17.03	0.8096
EBV DNA copies	≤4000	references			
	>4000	1.49	0.30	7.44	0.6273
Digital NPC-TILs		1.58	1.13	2.19	<0.05

**Table 4 cancers-15-05789-t004:** Multivariate analysis of risk factors on LRFS.

Covariate	Sub-Covariate	HR	Lower HR 95%	Upper HR 95%	*p*-Values
Age		1.05	0.96	1.14	0.285
Gender	Female	references			
	Male	1.20	0.15	9.62	0.866
T	1	references			
	2	0.47	0.01	30.02	0.7200
	3	2.78	0.21	36.08	0.4352
	4	1.20	0.06	24.18	0.9073
N	0	references			
	1	0.56	0.06	5.2	0.6107
	2	0.38	0.04	4.13	0.4290
	3	0.71	0.03	16.7	0.8291
Stage	I	references			
	II	0.44	0	88.01	0.7593
	III	0.96	0.05	17.89	0.9780
	IV	1.55	0.08	30.61	0.7723
EBV DNA copies	≤4000	references			
	>4000	1.04	0.17	6.34	0.963
Digital NPC-TILs		1.59	1.11	2.28	<0.05

## Data Availability

Code for generating digital NPC-TIL score can be accessed at https://github.com/mdsatria/npc_digital_tils (accessed on 21 November 2023) (will be available upon publication). Data can be made available upon request to the corresponding author (n.m.rajpoot@warwick.ac.uk).

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
