# Peer review of "AI-Based Risk Score from Tumour-Infiltrating Lymphocyte Predicts Locoregional-Free Survival in Nasopharyngeal Carcinoma"

_cancers, 2023, doi:10.3390/cancers15245789_

Round 1
Reviewer 1 Report
Comments and Suggestions for Authors
General/major comments
A. A general description of the work-flow is wanted: labour intensity, time consumption etc., particularly as an introduction into clinical practice is suggested.
B. A key focus of the paper is lymphocyte areas in NPC: how do the authors know that these cells are lymphocytes (in the absence of any use of immunohistochemistry)?
C. The discussion lacks in width and depth. Please, see some suggestions below. Also, why is not a comparison between stroma and cancer areas discussed?
Further comments
1. Line 20: the text should read: “This study aims to generate…”
2. Line 25: You consider plasma EBV-DNA as “the current main prognostic marker” for NPC. I suggest that Stage is the main prognostic marker and, therefore, the basis for stratification. What is your opinion on their relative importance?
3. When indicating EBV-DNA in the text, e.g., line 26, consider clarifying that it is plasma EBV-DNA that is discussed.
4. Line 29: the wording “tumour and lymphocyte nuclei” is awkward. Do you mean “cancer cell and lymphocyte nuclei”?
5. Line 32: the text should read: “… in a validation set…”
6. Line 33: “This finding also found…”. Do you mean “This finding was also found…”?
7. Line 38 (and line 346 + 352): somewhere in this ms. I would like to see a discussion/speculation on how this suggestion may assist in treatment decisions. Specifically, what changes to current regimes do the authors envisage? Introduced as an RCT? Treatment escalation/de-escalation? New treatments?
8. Line 45: to me, the Fossa of Rosenmüller is located in the (naso)pharynx, not in the nasal cavity.
9. Line 49: please, define “advanced cases”.
10. Line 60: Please, describe what sensitivity is referred to in the statement “EBV DNA is not as sensitive to locoregional recurrence as it is to distance metastasis”, and provide references.
11. Line 61-62: the authors state that “as much as 40% of local recurrence is not associated with an increase in plasma EBV DNA”. Please, provide additional references on the matter.
12. Line 65: Remove “the” in “the NPC”.
13. Line 71 (and line 314): please, consider including/using the publication by Nilsson et al. entitled “Immune phenotypes of nasopharyngeal cancer” (Cancers 2020; 12: 3428.)
14. Line 90: “locoregional recurrence (LRFS)”. Please, consider “locoregional recurrence free (LRFS)”.
15. Line 96-97: why were multiple sections used for some patients?
16. Line 104: why were “M” (distant metastases”) not included in the clinical data? Is this affecting your analysis against clinical parameters?
17. Line 115: how was the threshold of 2,000 copies / mL chosen?
18. Line 117: consider restaging from the 7th to 8th version of the UICC + AJCC staging system.
19. Line 124: please, expand on “digital TILs quantification”. Did you experience any difference in discriminating TILs from other cells, notably cancer cells?
20. Line 140-141. “… inner tumour cluster, while the expanded region was designated as the outer tumour cluster.” Do you use these terms elsewhere in the manuscript? For example, they do not appear in Figure 1.
21. Line 161: I may be wrong, but I cannot find the Table S6 that you refer to. If missing, please, correct this. (And, check carefully indications to figures/tables in general.)
22. Line 171: here, you refer to “clinicopathological” data and include TNM. Was M assessed? (Please, see also comment 16 above.)
23. Line 188: p-value (typo in my version of the text).
24. Line 243: please, clarify: … “the number of positive lymphnodes N=3 …
25. Line 267: Figure 3 is cropped at the top: the titles are not visible.
26. Line 273: Consider using “cancer cell nuclei” instead of “tumour nuclei”.
27. Line 284: Correct language: “larger tumour nuclei larger”.
28. Line 293: Figure 5 is cropped at the top: the titles are not visible.
29. Line 314: “With current limitations of the TNM staging system and clinical data [38,39] …,”. Reference 38 does not assess whether or not there are limitations to the TNM staging system, rather it deals with the tumour microenvironment of NPC. (So, it fits very well with the 2nd part of the sentence.)
30. Line 328-330: two subsequent sentences that state the same opinion. Please, consider revision of the text.
31. Line 336: “We found that TILs Density Cluster is the most common feature in digital NPC-TILs with highest importance across all survival endpoints.” Yet, in Figure 6, stromal TIL density tumour and stromal TILS ratio are listed as the top-ranked features. Please clarify this discrepancy.
32. Line: 352: Consider another wording for “rolled out” ´, e.g., transferred or implemented.
Comments on the Quality of English LanguageMy opinion is that the text would benefit from a revision by a native speaker. Some changes are suggested above, but there are further issues such as the annoying use of multiple tenses inside sentences/paragraphs.
Reviewer 2 Report
Comments and Suggestions for Authors
The present work, "AI-based Risk Score from Tumor Infiltrating Lymphocyte Predicts Locoregional Free Survival in Nasopharyngeal Carcinoma", presents a novel AI-based method to predict locoregional free survival (LRFS) in patients with nasopharyngeal carcinoma (NPC) using tumor infiltrating lymphocytes (TILs) quantified from hematoxylin and eosin (H&E)-stained slide images. The research idea is good. However, there are several issues that need authors revise.
1. The study is based on data from a single institution (Sun Yat-sen University Cancer Center). This raises concerns about the generalizability of the findings. How well can the results be applied to broader, more diverse populations?
2. The data collection process is well-detailed, but the justification for discarding 11 cases due to lack of associated clinical data should be elaborated.
3. The work should consider a wider variety of clinical characteristics such as lifestyle choices (e.g., smoking status, alcohol consumption), comorbidities (e.g., diabetes, hypertension), and other relevant medical histories that could influence nasopharyngeal carcinoma prognosis. The work should consider the variations in treatment protocols, such as different chemotherapy regimens, radiation therapy approaches, and surgical techniques. These factors can significantly impact patient outcomes and should be factored into the model's analysis.
4. Detailed information on the AI model architecture, training, and validation process is crucial but seems insufficiently addressed. It is crucial for the authors to describe why was ResNet chosen, what are its advantages in this specific application, and how the ResNet architecture is utilized, especially if it forms a core component of their AI model.
5. How was the model validated to ensure its reliability and accuracy, especially given the potential variability in H&E stained slide images?
6. The manuscript should detail how the model was validated. This includes the methods used to test the model's performance, such as cross-validation techniques, and how it handles potential overfitting.
7. Clarification on how misclassification of nuclei was addressed and the impact of this correction on the overall model performance is needed
8. Providing a comparison with literature such as the HoverNet model would help in understanding the novelty and effectiveness of the model used in this study.
9. Abstract line 37 needs to write LRFS full name.
10. Line 136-137 “After tumor clusters were generated, we expanded their area by 60 pixels or 15 microns.” The sentence needs rewriting.
11. The interpretability of the RSF model is crucial, especially in a clinical context. The manuscript should expand on how the identified features contribute to the model's predictions and their implications for clinical decision-making
Comments on the Quality of English LanguageMinor editing of English language required
Reviewer 3 Report
Comments and Suggestions for Authors
In this study, the authors developed a TILs score system by using the deep learning algorithm and proved the novel digital TILs score has prognostic value in terms of LRFS as well as other survival endpoints. These pathological markers could help clinicians with decision-making. This paper includes valuable researches which will be of interest for other research groups.However, there are several major limitations in this study that need to be addressed and these limitations may impact the robustness of the results.
1. Introduction
Please indicate and strengthen the clinical value of this research at the end of the Introduction (line93) to make this article more comprehensive.
2. Materials and Methods
1) Why you choose 2,000 copies/ml but not 4,000 copies/ml as a threshold to stratify NPC cases (line115), because more latest authoritative studies used 4,000 copies/ml as an effective threshold (e.g. doi: 10.1001/jama.2022.13997; doi: 10.1200/JCO.21.01467). Are there any other explanations?
2) In this study, only sex, TNM stage and EBV DNA copies were included in the clinicopathological features. Nevertheless, smoking history and nonkeratinizing histology may also have influence on the patients’ prognosis. You can take more clinical factors into consideration.
3) In this study, the authors term the tumours and TILs nuclei located in the expanded region (60 pixels or 15 microns) as stromal tumour cells and stromal (line151-152). However, in previous researches (e.g. doi: 10.1136/jitc-2019-000334) except for tumour cells the remaining tissue areas were classified as stroma area. Therefore, I think the expanded region could be more precisely described as tumor-related stromal or tumor invasive margin.
3. Results
1) In this study, the authors developed four sets of features in NPC prognosis, however, why not combined clinic-pathological features with NPC-TILs features together to predict survival event. Because in all feature sets, the TM features may influence the prediction performance of other features.
2) The authors didn’t show the formula of TILs score and the prognostic cut-off value, and all of those results will be valuable in application and reference.
3) ‘Univariate and multivariate analyse’. In routine analysis, the multivariate analysis controlled for all factors with significant associations emerging from the univariate analysis. However, the authors employed all features in the univariate analysis as well as multivariate analysis, please give reasons for such statistical method.
4) The title of Table 3 and 4 are not appreciate. ‘Univariate/ multivariate analysis of risk factors on LRFS’ may be better.
5) In Fig 4, the authors show the research results with visualization display, but in the subsequent analyses, the top-3 most important features in LRFS were Stromal TILs Density Tumour, Stromal TILs Ratio and TILs Density Cluster, and the current figure can’t display stromal TILs clearly. I think more visualization display, especially for stromal, will make readers understand your research better.
4. Discussion
1) In this study, the authors established a newly TILs score system with 12 digital TILs features, therefore how these 12 features were decided and more explanations of the results are required to help the reader with understanding the findings. The current explanation (line338-341) is too simple.
2) In line 160, the authors indicated the TM features were computed for downstream analysis, but in discussion part you didn’t explain the correlation between TM and TILs features. Please take it into consideration and make a new expression.
Comments on the Quality of English LanguageMinor editing of English language required.
Round 2
Reviewer 2 Report
Comments and Suggestions for Authors
No more suggestions
Reviewer 3 Report
Comments and Suggestions for Authors
I think this manuscript has met the approval to be published. You have developed an interesting research and I expect to more valuable works from your team.